# OpenReview forum: "Predicting the Performance of Black-box Language Models with Follow-up Queries"
_NeurIPS.cc/2025/Conference — NeurIPS 2025 poster_

### Official Review · Reviewer_ZuoB · 2025-06-26

**Clarity:** 3
**Significance:** 3
**Originality:** 3
**Rating:** 4
**Confidence:** 5

**Summary:**

This paper defines the task of black-box language model performance prediction and proposes a method (QueRE) based on follow-up queries to elicit predictive features from model outputs for model behavior prediction (e.g., truthfulness of generated responses). This approach is evaluated on three applications: predicting instance-level correctness in open/closed-ended QA tasks, detecting potential adversarial prompts, and distinguishing between black-box LLMs. QueRE performs competitively, often matching or outperforming white-box baselines.

**Questions:**

Please refer to Strengths And Weaknesses.

**Ethical Concerns:**

["NO or VERY MINOR ethics concerns only"]

**Final Justification:**

The authors have provided additional explanations that clarify the significance of the proposed new task and its contribution to the research community. The added experiments during the rebuttal phase further strengthen the completeness, soundness, and empirical support of the work and make me recognize the authors' viewpoint that the key finding of this work lies in revealing black-box model auditing as a potentially valuable research direction. These clarifications and new results have addressed my main concerns so I've changed my rating to borderline accept.

**Limitations:**

The method’s reliance on multiple follow-up queries per instance could increase inference cost in deployment settings.

**Paper Formatting Concerns:**

I did not notice any major formatting issues.

**Quality:**

3

**Strengths And Weaknesses:**

**Pros:**
1. The paper formally introduces a practically relevant task, especially since many mainstream LLMs are only accessible in black-box settings through APIs.
2. The authors design various evaluation settings and present extensive quantitative results. Some interesting applications are included, such as black-box model identification and performance prediction.
3. QueRE generally performs competitively even compared with white-box baselines.
4. The paper is clearly written and easy to understand.

**Cons:**
1. The technical contribution of the proposed QueRE is limited, as it relies heavily on LLM-generated prompts (i.e., follow-up questions) and an overly strong hypothesis that the predictive distributions for these questions meaningfully vary with correctness.
2. The significance of this task is closely tied to the achievable performance upper bound. I think the authors should further investigate whether the task of black-box performance prediction is in fact predictable—i.e., whether LLMs are sufficiently aware of their own factually incorrect outputs, and whether this internal perception can be effectively extracted via follow-up questions. If there is not enough headroom in performance, further exploration in this area may have limited practical utility.
3. **Regarding experimental settings:**
- In the model performance prediction task, the authors should additionally report precision and F1-score on negative samples (i.e., cases where LLMs produce incorrect answers), as higher precision indicates more reliable detection of incorrect behavior.
- A more realistic setup for black-box model identification should be determining whether a given response is or is not from a specific model. This implies that the negative samples in the test set should come from multiple different LLMs. The current setting—distinguishing between two specific models—is much easier and less representative of real-world applications, such as detecting fraudulent API substitutions.

---

> ### Author Rebuttal · Authors · 2025-07-31
>
> We thank the reviewer for their detailed comments. We appreciate your recognition of the clarity of our paper and the practical relevance of the black-box performance prediction task. We address your comments below:
>
> > **"The technical contribution of the proposed QueRE is limited, as it relies heavily on LLM-generated prompts (i.e., follow-up questions) and an overly strong hypothesis that the predictive distributions for these questions meaningfully vary with correctness."**
>
> While the mechanism of QueRE is intentionally simple, our core contribution lies in the empirical discovery and validation that this straightforward, non-obvious hypothesis holds true to a **surprisingly effective degree**. The key finding is not the hypothesis itself, but that such a simple black-box technique can yield representations powerful enough to consistently match, and even sometimes outperform, complex white-box methods that require privileged access to model internals. This result challenges the assumption that performant monitoring requires internal access and opens a new, practical avenue for black-box model auditing.
>
>
> > **"The significance of this task is closely tied to the achievable performance upper bound. I think the authors should further investigate whether the task of black-box performance prediction is in fact predictable—i.e., whether LLMs are sufficiently aware of their own factually incorrect outputs, and whether this internal perception can be effectively extracted via follow-up questions."**
>
> While measuring a theoretical performance upper bound is a challenging open question, our work provides the strongest evidence to date that a significant degree of a model's "internal perception" is indeed extractable via black-box queries. Standard uncertainty quantification methods perform poorly on these tasks, suggesting the problem is non-trivial. The fact that QueRE **significantly outperforms these methods** demonstrates that there is substantial headroom for improvement and that our approach is a major step toward that upper bound.
>
> From a usability perspective, this task is **practically relevant**. If we are able to accurately predict when a model will make a mistake, we can know when to use a model and when to defer to human intervention, a key goal for deploying LLMs in high-stakes settings.
>
>
> > **"In the model performance prediction task, the authors should additionally report precision and F1-score on negative samples (i.e., cases where LLMs produce incorrect answers), as higher precision indicates more reliable detection of incorrect behavior."**
>
> Thank you for this suggestion. We have added precision and F1 scores on negative (incorrect) examples for LLaMA3-8B and LLaMA3-70B. QueRE remains the strongest black-box method across nearly all tasks. Below are results in the form of precision / F1:
>
>
> | Dataset        | LLM           | Pre-conf    | Post-conf   | Answer P.   | Sem. Entropy | QueRE       |
> | -------------- | ------------- | ----------- | ----------- | ----------- | ------------ | ----------- |
> | **BoolQ**      | LLaMA3-8B     | 0.324/0.435 | 0.307/0.440 | 0.266/0.317 |     0.334/0.462         | 0.446/0.569 |
> | | LLaMA3-70B    | 0.410/0.509 | 0.410/0.529 | 0.325/0.362 | 0.427/0.550 | 0.591/0.684 |
> | **CS QA**      | LLaMA3-8B     | 0.804/0.567 | 0.821/0.553 | 0.857/0.609 |    0.970/0.485          | 0.920/0.836 |
> |                | LLaMA3-70B    | 0.807/0.765 | 0.843/0.710 | 0.817/0.735 |            0.891/0.928  | 0.953/0.943 |
> | **HaluEval**   | LLaMA3-8B     | 0.741/0.551 | 0.711/0.635 | 0.761/0.723 |  0.712/0.817            | 0.803/0.798 |
> |                | LLaMA3-70B    | 0.772/0.632 | 0.775/0.787 | 0.787/0.680 |            0.794/0.675  | 0.810/0.800 |
> | **DHate**      | LLaMA3-8B     | 0.374/0.411 | 0.508/0.540 | 0.373/0.516 |            0.444/0.546  | 0.747/0.761 |
> |                | LLaMA3-70B    | 0.394/0.380 | 0.456/0.437 | 0.360/0.506 | 0.491/0.433             | 0.785/0.777 |
>
>
> We will include these results in our revision.
>
> > **"A more realistic setup for black-box model identification should be determining whether a given response is or is not from a specific model. This implies that the negative samples in the test set should come from multiple different LLMs. The current setting—distinguishing between two specific models—is much easier and less representative of real-world applications, such as detecting fraudulent API substitutions."**
>
> Thank you for this suggestion. We completely agree that this is an important and realistic setting. We have now extended our evaluation to include multi-negative identification: determining whether a given response is from GPT-4o-mini, where negatives come from a pool of 3 other models (LLaMA3-8B, LLaMA3-70B, GPT-3.5). QueRE remains highly effective, achieving near-perfect performance and demonstrating its value for real-world applications like detecting fraudulent API substitutions. We will include these results in our revision.
>
> |Dataset|Pre-Conf|Post-Conf|Answer Probs|Sem. Entropy|QueRE|
> |-|-|-|-|-|-|
> |SQuAD| 0.541| 0.522| 0.501 |0.5798| **0.998**|
> |BoolQ| 0.501| 0.532| 0.564 |0.5921| **0.998**|
>
> Thank you again for your detailed review, and we hope our new experiments and our clarifications address your concerns.

---

### Official Review · Reviewer_hVT7 · 2025-06-30

**Clarity:** 3
**Significance:** 3
**Originality:** 3
**Rating:** 4
**Confidence:** 3

**Summary:**

This paper proposes QueRE, a black-box method of predicting model performance as well as discriminating models. The method is then compared to baselines, both white-box and black-box, on a variety of tasks ranging from predicting model performance on closed and open-ended question answering to distinguishing between different models.

**Questions:**

1) In L117, how is the distribution over the set of possible answers calculated in the case of LLMs that don’t return probabilities? Also, I think the sentence is grammatically incorrect. Either the distribution over possible answers or what?
2) As mentioned in weaknesses point 1, the datasets in section 4 are mostly released before the models evaluated have been released. Do the authors think this would generalize to newer datasets that are definitely not in the training set of these models?
3) Would it be possible to measure the ability of this approach to differentiate between a full/half precision model and a more severely quantized (<=4bit) version of the model? I think this is another important usecase for evaluating serving APIs aside from checking if they are serving the right model.
4) Minor detail for final version: The code has a typo with the name of the Boolean Questions (BoolQ) dataset, using i instead of an L for the dataset name.

**Ethical Concerns:**

["NO or VERY MINOR ethics concerns only"]

**Final Justification:**

The authors adequately address my points, I have no further comments and would like to maintain my score.

**Limitations:**

yes

**Paper Formatting Concerns:**

No formatting issues

**Quality:**

4

**Strengths And Weaknesses:**

**Strengths**:
1) Simple approach that is not tailored to a specific model
2) Strong performance and generalization compared to other methods
3) Useful and practical approach that is directly applicable
4) Experiments on many datasets and different tasks

**Weaknesses**:
1) Lack of datasets that are not potentially contaminated (As far as I could see, HaluEval and GPT3.5 is the only combination that is not potentially contaminated, assuming that later versions of GPT3.5, namely GPT3.5-Turbo-0125, have not been further trained).
2) The choice of using pre- and post- confidence scores in addition to the original vector z in the case of QA tasks should be ablated.

---

> ### Author Rebuttal · Authors · 2025-07-31
>
> We thank the reviewer for their detailed and positive feedback and for their excellent suggestions. We are encouraged that they found our approach to be simple, strong, and practical. We address the questions and suggestions below.
>
> > **"As mentioned in weaknesses point 1, the datasets in section 4 are mostly released before the models evaluated have been released. Do the authors think this would generalize to newer datasets that are definitely not in the training set of these models?**
>
> Yes, we believe our method generalizes well to evaluation datasets that are not part of the model's pretraining distribution. To support this, in addition to GPT-3.5 on HaluEval (as you note), we include results for LLaMA2-7B and LLaMA2-70B on HaluEval, a dataset released after the training of LLaMA2. We observe that QueRE continues to perform strongly:
>
> | |Pre-conf|Post-conf|Semantic Ent.|QueRE|
> |:-|-:|-:|-:|-:|
> |LLaMA2-7B|0.5013|0.6647|0.5833|0.7819|
> |LLaMA2-70B|0.5237|0.5399|0.5223|0.6935|
>
> This suggests that performance remains strong on evaluations that are guaranteed to not be contained in the training data.
>
> > **"The choice of using pre- and post- confidence scores in addition to the original vector z in the case of QA tasks should be ablated."**
>
> Thank you for the suggestion. We conducted an ablation in which we removed the pre- and post-confidence scores and found that the performance of QueRE remains strong. Notably, even without these features, QueRE continues to outperform the next-best black-box method (*Semantic Entropy*). Below are results for LLaMA3-70B:
>
> ||BoolQ|CS QA|HaluEval|DHate|WinoGrande|
> |-|-|-|-|-|-|
> |Sem. Entropy|0.787|0.898 |0.543 |0.621 | 0.528|
> |QueRE without pre-, post-conf| 0.861 | 0.963 |0.789| 0.599|0.541|
> |QueRE|0.901 |0.964 |0.790|0.600|0.545|
>
> We will include these results in our revision.
>
> > **"In L117, how is the distribution over the set of possible answers calculated in the case of LLMs that don’t return probabilities? Also, I think the sentence is grammatically incorrect. Either the distribution over possible answers or what?"**
>
> In our experiments, we use the top-5 logits returned by the model to compute the distribution over possible answers. If a candidate answer appears in the top-5, we normalize its logit over the total mass of the top-5 to estimate its probability; otherwise, it is assigned probability 0.
>
> Thank you for catching the grammatical issue. The intended sentence is: "we append the distribution over possible answers." We will correct this in our revision.
>
> > **"Would it be possible to measure the ability of this approach to differentiate between a full/half precision model and a more severely quantized (<=4bit) version of the model? I think this is another important usecase for evaluating serving APIs aside from checking if they are serving the right model."**
>
> This is an excellent suggestion for another practical use case. We have run this experiment, extending our work on distinguishing between models. We find that QueRE can accurately distinguish between full-precision, half-precision, and 4-bit quantized versions of LLaMA3 models on the SQuAD dataset. These promising results suggest QueRE can be a valuable tool for auditing APIs to detect if providers are serving lower-precision models.
>
> ||Pre-conf|Post-conf|Logprobs|Semantic Ent.|QueRE|
> |-|-|-|-|-|-|
> |LLaMA3-3b (4bit v 32bit)|0.61|0.59|0.57|0.71|**0.99**|
> |LLaMA3-3b (16bit v 32bit)|0.50|0.51|0.51|0.58|**0.99**|
> |LLaMA3-8b (4bit v 32bit)|0.67|0.50|0.61|0.63|**0.98**|
> |LLaMA3-8b (16bit v 32bit)|0.51|0.54|0.55|0.56|**0.97**|
>
>
> > **"The code has a typo with the name of the Boolean Questions (BoolQ) dataset, using i instead of an L for the dataset name."**
>
> Thank you for catching this! We have fixed this typo in our code and will ensure the correction appears in the final version.

---

### Official Review · Reviewer_Sso9 · 2025-07-01

**Clarity:** 3
**Significance:** 4
**Originality:** 4
**Rating:** 5
**Confidence:** 5

**Summary:**

This paper introduces an approach for predicting the behavior of black-box language models by asking them a series of follow-up questions. The approach, QueRE, works by taking the probabilities of "Yes" responses to these questions and using them as features to train a simple linear predictor. The research demonstrates that this black-box technique can reliably and accurately predict a model's correctness on various question-answering and reasoning benchmarks, surprisingly outperforming even white-box methods that have access to the model's internal activations. Furthermore, the paper shows that these features can be used to effectively detect when a model has been adversarially influenced by a system prompt (e.g., to answer incorrectly or generate buggy code) and to accurately distinguish between different black-box LLMs (e.g., GPT-3.5 vs. GPT-4o-mini), offering a promising direction for monitoring the reliability and integrity of closed-source models.

**Questions:**

1. The approach shows surprisingly strong performance in predicting correctness on reasoning-intensive tasks like GSM8K and CodeContests, where many other methods fail.  Given that the follow-up questions seem to probe for general confidence rather than specific reasoning steps, do the authors have a hypothesis for why these features are so predictive for complex reasoning tasks?
2. The ablation study in Figure 5 shows that performance increases with the number of elicitation questions.  Have the authors investigated methods for optimizing this set of questions to identify a minimal, yet highly informative, subset? This could help reduce the computational cost associated with the method.

**Ethical Concerns:**

["NO or VERY MINOR ethics concerns only"]

**Final Justification:**

The rebuttal strengthens my recommendation of accept for this paper. The framework is simple, does not require access to model internals and provides a strong performance. This paper will certainly benefit the community.

**Limitations:**

Yes

**Quality:**

3

**Strengths And Weaknesses:**

The paper has the following key strengths:
- A simple yet highly effective approach: The proposed approach does not require access to model internals to calibrate the confidence but is remarkably effective, often matching or even outperforming white-box methods that require access to the model's internal hidden states or full logits.
- Versatility across multiple critical tasks: The utility of the features extracted by QueRE is demonstrated across three distinct and important applications. It can accurately predict instance-level correctness on a wide range of QA and reasoning benchmarks, reliably detect when an LLM has been adversarially influenced by a system prompt, and accurately distinguish between different black-box LLMs, which is useful for auditing APIs.
- Strong generalization and transferability: The predictors trained using QueRE show strong generalization properties and perform well on out-of-distribution data. Experiments demonstrate that the predictors transfer effectively across different QA datasets (for example, from SQUAD to NQ) and across different model scales (for example, from LLaMA3-3B to 8B) without needing new labeled data from the target task.

The paper has the following weaknesses:
- Computational cost overhead compared to model internals approach: The approach relies on asking n follow up questions (n = 50 shows good performance in the paper). The authors note in line 116 that these follow up questions can be asked in parallel adding minimal computational overhead. However, parallelizing the computation would reduce latency but the cost would still remain high. So for a given query, a model would have to be called 50 times to estimate the calibrated confidence which is quite expensive. Model internals approach as well as other baselines do not have such high computational costs. Also, since follow up questions are asked which rely on using the original output as an input, latency would also increase. I would suggest the authors to mention this explicitly in their paper for transparency.
- Results only shown on simpler tasks: The datasets considered in the paper are comparatively simpler for the LLMs shown in the paper. For example, on GSM-8K, GPT-4 has a performance of ~95%. Confidence calibration in such tasks would be a lot different than tasks on which the models have medium or low performance (such as Olympiad level math problems, multi-turn function calling). Having results on such difficult tasks can certainly strengthen the paper. I would also encourage authors to add model’s accuracy for each of the tasks so that this is also clearer.
- Comparison to stronger baselines like MICE: The authors have included white box and black box baselines for comparison. I would also encourage the authors to include stronger baselines such as [1].

[1] Subramani, Nishant, et al. "MICE for CATs: Model-Internal Confidence Estimation for Calibrating Agents with Tools." Proceedings of the 2025 Conference of the Nations of the Americas Chapter of the Association for Computational Linguistics: Human Language Technologies (Volume 1: Long Papers). 2025.

---

> ### Author Rebuttal · Authors · 2025-07-31
>
> We thank the reviewer for their positive feedback! We appreciate that you find our approach "simple yet highly effective" and that we tackle "multiple critical tasks". We address your comments below:
>
> > **"Computational cost overhead compared to model internals approach"**
>
> Thank you for pointing this out. While QueRE does require multiple forward passes---one per follow-up question---we note that this cost is a trade-off inherent to the black-box setting, where **model internals approaches are not applicable**. We will update the paper to explicitly discuss this trade-off and its implications for deployment. In many high-stakes applications (e.g., safety-critical monitoring, auditing), we believe the **improved predictive performance of QueRE** justifies this cost.
>
> > **"The authors note in line 116 that these follow up questions can be asked in parallel adding minimal computational overhead. However, parallelizing the computation would reduce latency but the cost would still remain high..., other baselines do not have such high computational costs."**
>
> We would like to highlight that QueRE is quite comparable in terms of **both computational cost and latency** to the other competitive black-box baselines of *Semantic Entropy* and *Self-consistency*. A key advantage of QueRE is that it only requires $k$ forward passes for the probabilities of a single token (e.g., "Yes") in response to $k$ follow-up questions.
>
> First, this can even be faster than generating $k$ full sequences of many tokens (each of which requires $N$ forward passes to generate $N$ tokens), which is required by both *Semantic Entropy* and *Self-consistency*. In addition to these generations, Semantic Entropy then requires an additional step **after** sampling of using an embedding model and then clustering the embeddings of the responses. Secondly, in terms of computational cost, APIs often charge for the number of output tokens, which is much higher for these baselines.
>
> To make this comparison more concrete, we have provided the results on SQuAD with LLaMA3-8B in terms of average runtime per example below. We find that our approach dominates other baselines at the same runtime. For example, QueRE with 10 follow-up questions outperforms Self-Consistency, which takes a similar amount of time, and strongly outperforms Semantic Entropy, which is much slower.
>
> || AUROC | Avg Runtime Per Example (seconds) |
> |-|-|-|
> |Post-conf|0.515| 0.08 |
> |QueRE w/ 5 followups | 0.868 | 0.13 |
> |QueRE w/ 10 followups | 0.897 | 0.17 |
> |QueRE w/ 20 followups| 0.916 | 0.36 |
> |QueRE w/ 30 followups| 0.928 | 0.55 |
> |QueRE w/ 40 followups| 0.933 | 0.74 |
> |QueRE w/ 50 followups| 0.949 | 0.89 |
> |Self-Consistency|0.534|0.19|
> |Semantic Entropy|0.521 |2.44 |
>
>
> > **"Results only shown on simpler tasks: The datasets considered in the paper are comparatively simpler for the LLMs shown in the paper... I would also encourage authors to add model’s accuracy for each of the tasks"**
>
> This is an excellent suggestion. To clarify that the benchmarks remain challenging for the models we evaluated, we have compiled the models' accuracies below. As the data shows, performance is far from saturated on these tasks, making performance prediction a non-trivial and valuable problem. We will add a similar table to the Appendix in our revision.
>
> |Model|BoolQ Acc|HaluEval Acc|DHate Acc|CS QA Acc|WinoGrande Acc|Squad Exact Match|NQ Acc|
> |-|-|-|-|-|-|-|-|
> |LLaMA3-3B|0.777|0.632|0.655|0.584|0.588|0.516|0.186|
> |LLaMA3-8B|0.799|0.712|0.673|0.603|0.602|0.545|0.302|
> |LLaMA3-70B|0.812|0.774|0.694|0.713|0.651|0.757|0.403|
> | GPT-3.5     | 0.874     | 0.772        | 0.780     | 0.792     | 0.670          | 0.584             | 0.449  |
> | GPT-4o-mini | 0.890     | 0.786        | 0.797     | 0.834     | 0.713          | 0.653             | 0.451  |
>
> ||GSM8K Acc|CodeContests Acc|
> |-|-|-|
> |GPT-3.5|0.794|0.392|
> |GPT-4o-mini|0.853|0.434|
>
>
> > **"Comparison to stronger baselines like MICE: The authors have included white box and black box baselines for comparison"**
>
> Thank you for pointing us to this interesting work. We wish to re-emphasize that our paper is focused on the black-box setting, where such white-box approaches are not applicable. We include white-box baselines like RepE and Full Logits not as direct competitors, but rather to demonstrate that QueRE can surprisingly match—or even exceed—their performance without access to model internals. Furthermore, the MICE method appears to be a strong white-box technique for confidence estimation. However, as of writing, the code for [1] has not been publicly released—their GitHub repository is currently empty—which prevents a direct comparison.
>
>
>
> > **"The approach shows surprisingly strong performance in predicting correctness on reasoning-intensive tasks like GSM8K and CodeContests, where many other methods fail. Given that the follow-up questions seem to probe for general confidence rather than specific reasoning steps, do the authors have a hypothesis for why these features are so predictive for complex reasoning tasks?"**
>
> This is a great question. While our follow-up questions are general, we hypothesize they are particularly effective on reasoning tasks (e.g., GSM8K, CodeContests) precisely because those tasks require long, structured outputs (i.e., chain-of-thought). After generating a complex reasoning chain, the model's internal state is likely much richer and more nuanced than after producing a single-token answer.
>
> When we probe this state with follow-up questions, the model’s response probabilities (e.g., P(“Yes”)) serve as rich signals—capturing subtle cues in its reasoning trajectory. In essence, the model has more context from its own reasoning to "reflect" upon, allowing our probes to better distinguish between confident (and likely correct) reasoning paths versus flawed ones. In contrast, simple QA tasks often involve short outputs (e.g., one-word answers), which provide less of a signal.
>
> > **"Have the authors investigated methods for optimizing this set of questions to identify a minimal, yet highly informative, subset? This could help reduce the computational cost associated with the method."**
>
> This is an excellent suggestion. As noted in lines 134–138, we have considered question optimization strategies (e.g., learning soft prompts), but opted for a fixed-question approach for several reasons: (1) it provides strong transferability across models and tasks, and (2) it enables tighter generalization bounds, since the questions are not optimized on data.. That said, we agree that identifying a minimal, highly informative question set is a promising direction, and we will highlight this as a valuable area for future work.

---

> > ### Comment · Reviewer_Sso9 · 2025-08-05
> >
> > Thanks for answering my questions. I maintain that this is a strong and interesting work and should be accepted.

---

### Official Review · Reviewer_fGiM · 2025-07-04

**Clarity:** 3
**Significance:** 3
**Originality:** 3
**Rating:** 5
**Confidence:** 4

**Summary:**

This paper aims to predict the behavior of black-box language models, e.g. predicting whether their outputs are correct. The paper proposes a method by asking follow-up questions and taking the probability of responses (e.g. the probability of "yes") as representations to train linear probes. Surprisingly, this simple method can outperform white-box linear method (e.g. train on the output logits).

**Questions:**

see weaknesses.

**Ethical Concerns:**

["NO or VERY MINOR ethics concerns only"]

**Final Justification:**

The rebuttal addresses all my concerns, especially about the latency and the robustness of the proposed method.
While the method is simple, this paper is conceptually interesting and empirically effective. I'd keep my score as accept.

**Limitations:**

Do you think there are any types of tasks that the proposed method might fail?

**Quality:**

3

**Strengths And Weaknesses:**

Strength
- This is an interesting new research problem. The demonstrated use case (e.g., judging output correctness, distinguishing adversarially perturbed LMs, and differentiating different LMs) justifies the importance of this problem setup.
- The main idea of asking follow-up questions is common, but to the best of my knowledge, this is the first paper that takes the probability of responses as representations to train linear probes.
- The experiment results are strong. QueRE consistently outperforms a wide range of strong baselines, including white-gox methods like RepE and Full Logits.

Weaknessess
- While QueRE gets strong performance on open-sourced models, it sometimes gets limited performance on closed-sourced models (e.g. Figure 2).
- The studied benchmarks (e.g. SQUAD, Natural Questions) are outdated. It'd be great to see experiments on more realistic benchmarks (e.g. coding, human preference).
- It's unclear how different follow-up questions might influence the performance of the proposed method. In particular, some follow-up questions are hand-specified, and the others are generated by prompting GPT-4.
- While the authors claim that the latency issue can be mitigated through batching, I didn't find any relevant reports about the computation overhead, especially with more follow-up questions (e.g. 100). Could you present the results of latency v.s. AuROC?

---

> ### Author Rebuttal · Authors · 2025-07-31
>
> We thank the reviewer for their positive feedback. We are glad to hear that you find our problem setting as "interesting" and that our "experiment results are strong". We address your individual comments below:
>
> > **"While QueRE gets strong performance on open-sourced models, it sometimes gets limited performance on closed-sourced models (e.g. Figure 2)"**
>
> We thank the reviewer for this careful observation. While QueRE's performance on closed-source models in Figure 2 (NQ, SQUAD) is competitive, we agree it is not *as dominant* as with open-source models on those specific tasks. However, we wish to highlight that across a broader range of evaluations, QueRE demonstrates consistently state-of-the-art performance for closed-source models. For instance:
>
> * On reasoning benchmarks (Table 1), QueRE is the top-performing method for both GPT-3.5 and GPT-4o-mini on GSM8K and CodeContests, significantly outperforming all baselines.
> * On closed-ended QA tasks (Figure 3), QueRE is the best black-box method for GPT models on HaluEval, BoolQ, and DHate.
> * In our other application settings, QueRE achieves near-perfect accuracy in detecting adversarially influenced GPT models (Table 2) and distinguishing between different GPT models (Figure 4).
>
> This comprehensive evidence suggests that our method is highly effective for closed-source models across a wide variety of important tasks beyond standard QA tasks.
>
> > **"The studied benchmarks (e.g. SQUAD, Natural Questions) are outdated. It'd be great to see experiments on more realistic benchmarks (e.g. coding, human preference)."**
>
> We appreciate this suggestion. While we included SQUAD and NQ as standard, widely-recognized QA benchmarks, we made a concerted effort to evaluate on more modern and diverse tasks to ensure the relevance of our findings. Our experiments include results on:
> * **Coding**: The CodeContests benchmark (Table 1).
> * **Mathematical Reasoning**: The GSM8K benchmark (Table 1).
> * **Recent Datasets**: The HaluEval benchmark, a recent and challenging hallucination detection dataset from 2023 (Figure 3).
>
> Our evaluation spans commonsense reasoning, factual recall, toxicity detection, and hallucination detection, in addition to the crucial applications of detecting adversarial tampering and auditing model APIs. We believe this diverse suite of benchmarks provides a robust validation of QueRE's capabilities.
>
> > **"It's unclear how different follow-up questions might influence the performance of the proposed method"**
>
> This is an excellent question. We investigated this in our ablation study in Appendix A.4 and Figure 7. In this study, we prompted GPT-4o to generate sets of questions with varying levels of human-interpretable diversity ("similar" vs. "diverse" questions). Our findings indicate that while performance steadily increases with the number of questions, the specific generation strategy (e.g., prompting for diversity) does not have a major impact when a sufficient number of questions are used.  This suggests our method is robust to the exact composition of the follow-up query set. We will emphasize this analysis in our revision.
>
> > **"While the authors claim that the latency issue can be mitigated through batching, I didn't find any relevant reports about the computation overhead... Could you present the results of latency v.s. AuROC?"**
>
> We have now included latency-vs-performance results for LLaMA3-8B on SQuAD, varying the number of follow-up questions up to the full 50 questions we use in our experiments. The following table compares QueRE to the key black-box baselines:
>
> || AUROC | Avg Runtime Per Example (seconds) |
> |-|-|-|
> |Post-conf|0.515| 0.08 |
> |QueRE w/ 5 followups | 0.868 | 0.17 |
> |QueRE w/ 10 followups | 0.897 | 0.17 |
> |QueRE w/ 20 followups| 0.916 | 0.36 |
> |QueRE w/ 30 followups| 0.928 | 0.55 |
> |QueRE w/ 40 followups| 0.933 | 0.74 |
> |QueRE w/ 50 followups| 0.949 | 0.89 |
> |Self-Consistency|0.534|0.19|
> |Semantic Entropy|0.521 |2.44 |
>
> We find that QueRE consistently outperforms other approaches at similar or lower runtimes. The results clearly show that QueRE offers a superior trade-off. For a comparable latency to Self-Consistency (0.17s vs. 0.19s), QueRE with just 5-10 follow-ups achieves a dramatically higher AUROC (~0.90 vs. 0.53). Furthermore, QueRE significantly outperforms the much slower Semantic Entropy baseline. We will include this latency analysis in our revision.
>
> > **"Do you think there are any types of tasks that the proposed method might fail?"**
>
> This is a great question! Our approach is designed to create a low-dimensional representation that is highly predictive of an LLM's behavioral properties (e.g., correctness, truthfulness, model origin). We believe this representation would be less suitable for fine-grained tasks, such as generation. For example, it would likely not be effective for predicting the exact sequence of tokens in a long-form text generation task.
>
> Its strength lies in classification and regression tasks related to the model's state and the quality of its output. We believe this is a reasonable and important scope, as these representations are shown to be highly effective for crucial applications like **monitoring model performance** and **detecting adversarial influence**. We will add this clarification to our revision.

---

> > ### Comment · Reviewer_fGiM · 2025-08-06
> >
> > Thanks for the detailed reply. My main concerns are all resolved and I'll keep my score.

---

### Decision · Program_Chairs · 2025-09-17

**Decision:**

Accept (poster)

**Comment:**

This paper proposed a method (QueRE) to predict the output quality of an LLM without tapping into its weights (aka as a blackbox). It asks the model-under-test follow up questions that expect binary answers, and use the probability of the answer (yes or no) as predictors to estimate the model's performance. This is different from other methods that uses the activation as the predictor (aka whitebox).

Paper strength.

The method is creative to leverage the multi-turn capability of LLMs.
It is simple that no complicated weight computation or access to internal state is needed, just yes no questions.
Another benefit of being simple is that it can generalize to different models and tasks.
The method works surprisingly well, backed by comprehensive experiments on different datasets and tasks.

Paper weakness.

Most of the concerns have been adequately addressed by the rebuttal, thus won't be repeated here. One reviewer questioned if the performance would hit an upper bound due to the model's limited (if so) true self-awareness of its own error. This is more like a future research direction. Another practical weakness is the computational cost of generating and sampling mutli-turn answers. The authors provided additional experiments but overall this is a key part of the method that is not easy to speed up. Another reviewer suggested the comparison to a strong whitebox baseline like MICE, which the authors acknowledged but won't be able to do immediately due to lack of open source code.

Reviewers also suggest the task of identifying the source model behind an API, which the authors provide new evaluations and promises to include in the final revision.

AC recommends acceptance of the paper given its novelty, simplicity and effectiveness.